# Spatiotemporal Assessment of Soil Organic Carbon Change Using Machine-Learning in Arid Regions

Hassan Fathizad [1,*], Ruhollah Taghizadeh-Mehrjardi [2,3,4,*], Mohammad Ali Hakimzadeh Ardakani [1], Mojtaba Zeraatpisheh [5,6], Brandon Heung [7] and Thomas Scholten [2,3,4]

[1]  Department of Arid and Desert Regions Management, School of Natural Resources & Desert Studies, Yazd University, Yazd 89195741, Iran; hakim@yazd.ac.ir

[2]  Department of Geosciences, Soil Science and Geomorphology, University of Tübingen, 72070 Tuebingen, Germany; thomas.scholten@uni-tuebingen.de

[3]  CRC 1070 Resource Cultures, University of Tübingen, Gartenstraße 29, 72070 Tuebingen, Germany

[4]  DFG Cluster of Excellence "Machine Learning", University of Tübingen, 72074 Tuebingen, Germany

[5]  Henan Key Laboratory of Earth System Observation and Modeling, Henan University, Kaifeng 475004, China; mojtaba.zeraatpisheh@henu.edu.cn

[6]  College of Geography and Environmental Science, Henan University, Kaifeng 475004, China

[7]  Department of Plant, Food, and Environmental Sciences, Faculty of Agriculture, Dalhousie University, Halifax, NS B3H 4R2, Canada; brandon.heung@dal.ca

*  Correspondence: h.fathizad@stu.yazd.ac.ir (H.F.); ruhollah.taghizadeh-mehrjardi@mnf.uni-tuebingen.de (R.T.-M.)

**Abstract:** Soil organic carbon (SOC) is an essential property of soil, and understanding its spatial patterns is critical to understanding vegetation management, soil degradation, and environmental issues. This study applies a framework using remote sensing data and digital soil mapping techniques to examine the spatiotemporal dynamics of SOC for the Yazd-Ardakan Plain, Iran, from 1986 to 2016. Here, a conditioned Latin hypercube sampling method was used to select 201 sampling sites. A set of 37 environmental predictors were obtained from Landsat imagery taken in 1986, 1999, 2010 and 2016. Here, SOC was modeled for 2016 using the Random Forest (RF), support vector regression (SVR), and artificial neural networks (ANN) machine-learners by correlating environmental predictors with soil data. The results showed that RF yielded the highest accuracy ($R^2 = 0.53$), compared to the other two learners. By performing a variable importance analysis of the RF model, normalized difference vegetation index, modified vegetation index, and ground-adjusted vegetation index were determined to be the most important environmental predictors. By applying the model calibrated from 2016 data to 1986, 1999 and 2010, the results showed a substantial decrease in SOC; these decreases in SOC were mainly attributed to land use changes and agricultural activities.

**Keywords:** random forest; machine learning; spatial distribution; variable importance analysis; vegetation index; temporal change

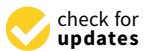



## 1. Introduction

Soil organic carbon (SOC) has a significant impact on many soil functions, such as the production of food and other biomass; and the provisioning of biological habitats and genetic resources. It is an important indicator for assessing and managing soil fertility, soil quality, and soil degradation [1,2]; hence, accurate information on the spatiotemporal patterns of SOC are required to support sustainable land use and management. Furthermore, information on the spatiotemporal variability of SOC is particularly important in the context of climate change at the local- [3], regional- [4], and global-scales [5]. As a result, methods for measuring, modelling, and monitoring SOC are continuously evolving around the world; however, methods of direct sampling and soil analysis using laboratory or field measurements are generally expensive and time-consuming to perform and therefore impractical to monitor the SOC changes over large spatial extents [6–8].

The recent development of digital soil mapping methods (DSM) provides a framework for characterizing the spatiotemporal patterns of soil properties [9]. DSM approaches involve the creation and operation of terrestrial spatial information systems obtained from field and laboratory observations, combined with spatial and non-spatial inference systems to generate raster-based map prediction and their respective uncertainty estimates [10–12]. These approaches apply statistical tools to quantify the relationships between soil properties and environmental predictors [8]. Compared to traditional approaches of soil mapping, DSM methods provide a more accurate representation of soil variability [7,11] and provide a suite of map products for supporting decision-making processes that are designed to address agricultural and other environmental issues [8]. When developing predictive models of SOC, previous studies have applied the use of machine learners, such as artificial neural networks (ANN), Cubist model trees [13,14], decision trees [15], support vector regression (SVR), and geostatistical approaches via kriging, inverse distance weighted interpolation, and spline approaches [16–18]. Additionally, the Random Forest (RF) learner has become increasingly popular in DSM research [13,19,20].

Many studies have predicted SOC using DSM techniques for a single point in time, but do not account for its temporal dynamics in models [6,13,21,22]. To address this issue, Fathizad et al. [23] provided a framework, which involved training and validating their predictive models using current data to map soils (i.e., soil salinity) and applied the model to historical environmental predictors acquired from remote sensing [23]. A similar approach has been applied in Taghizadeh-Mehrjardi et al. [24] to map historical patterns of heavy metal in soil samples from Iran [24].

The arid and semi-arid regions of Iran, as well as its deserts, are vulnerable ecosystems that are inhabited by large populations. These landscapes cover most of the terrestrial land-base of Iran; hence, methods for monitoring SOC in these areas are particularly important—especially given the presently low concentrations of SOC in these regions. Because there are considerable environmental challenges of conducting field studies in these areas of Iran, there is very little spatial information about soils and how they change over time. Hence, the objectives of this study were (1) to compare the performance of three ML models (RF, SVR, and ANN) to predict the spatial distribution of SOCs and (2) to study the changes of SOC from 1986 to 2016 in the Yazd-Ardakan plain.

## 2. Materials and Methods

The methodological framework that was used to carry out this study is summarized in Figure 1.

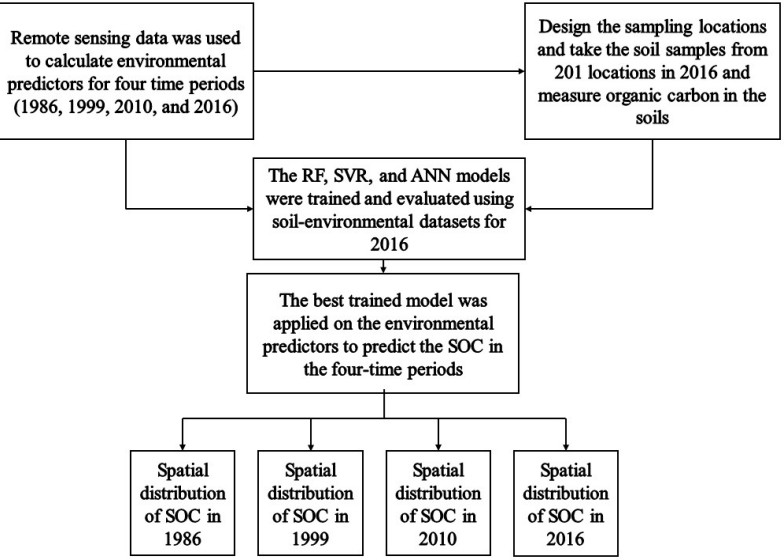

**Figure 1.** A methodological framework for predicting the spatiotemporal dynamics of soil organic carbon (SOC) using Random Forest (RF), support vector regression (SVR), and artificial neural networks (ANN).

### 2.1. Study Area & Soil Sampling

The Yazd-Ardakan Plain is located on the central plateau of Iran and in the central part of Yazd Province (53°45′14.6″ E to 54°48′18.1″ E longitude; and 31°51′0.6″ N to 32°26′37.5″ N longitude) and has an area of 4829 km². The study area has an elevation range of 977 m to 2684 m above mean sea level; the rainfall in this area is low and irregular (about 118 mm/year), and the evaporation rate ranges from 2200 to 3200 mm/year. According to the United States Classification of Soil Classification, the soil in the area is mainly described as Entisols and Aridisols soil. Geologically, the Yazd-Ardakan Plain is mainly composed of intrusive rocks, especially Neogene formations, and is covered by Quaternary alluvial deposits and agglomerations. The geological formations in the study area are varied with the youngest formations originating from the Quaternary period, which covers most of the territory.

This study used a conditioned Latin hypercube sampling method to select 201 sampling locations [25]. A global positioning system was used to find the geographic location of the sampling points, where soil samples were taken from the topsoil layer (i.e., 0–20 cm depth increment; Figure 2). SOC was measured using the wet oxidation (combustion) method [26].

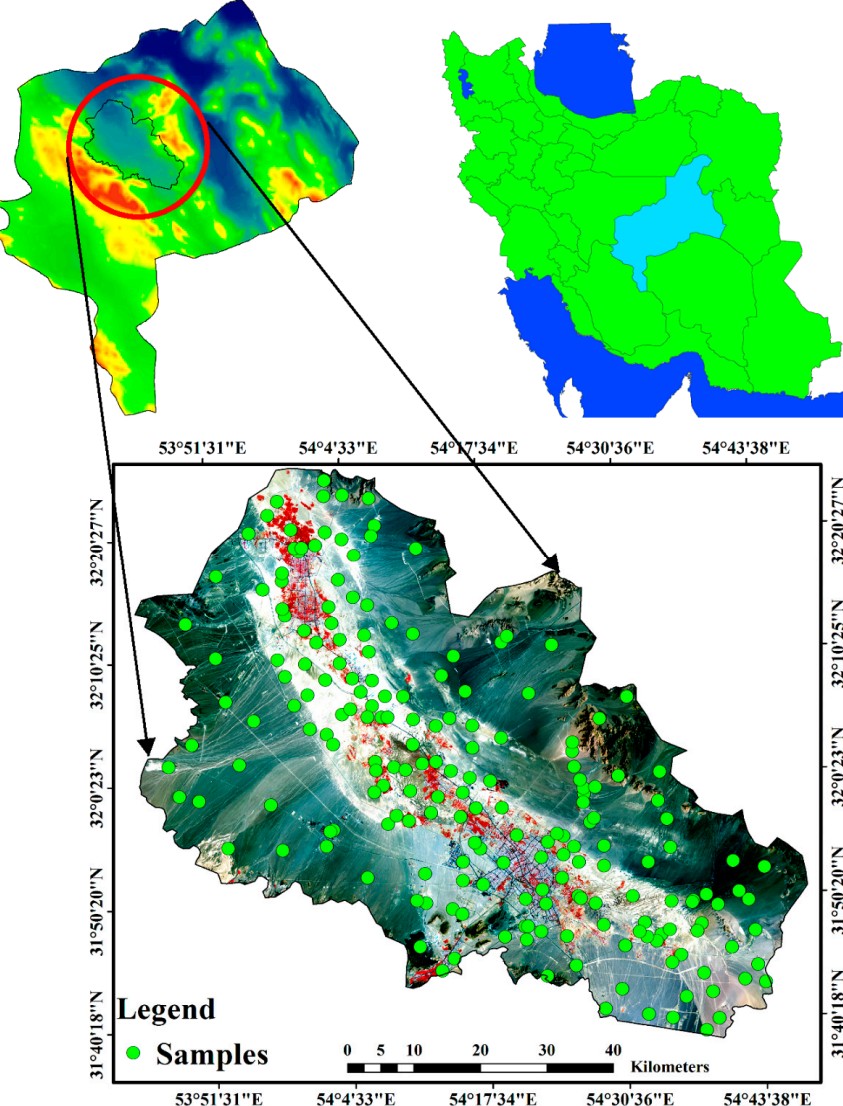

**Figure 2.** Location of the Yazd-Ardakan Plain and sampling points overlaid on a false color composite Landsat image.

## 2.2. Environmental Predictors

Satellite images obtained by Landsat 5 in 1986, Landsat 5 in 1999, Landsat 7 in 2010, and Landsat 8 in 2016 were used to calculate vegetation indices and other remote sensing indices. After carrying out preprocessing of the satellite images, a total of 37 vegetation indices were calculated. These indices included Normalized Differential Vegetation Index (NDVI), Soil Adjusted Vegetation Index (SAVI), Ratio Vegetation Index (RVI), Distinct Vegetation Index (DVI), and the Principal Components (PC) of the spectral bands. A complete list of the vegetation indices used in this study is presented in Table 1.

**Table 1.** List of environmental predictors derived from Landsat imagery.

| Covariates | Definition | Reference |
|---|---|---|
| Difference vegetation index (DVI) | NIR–Red | [27] |
| Enhanced vegetation index (EVI) | $G \times (NIR–Red)/(NIR + c1 \times Red–c2 \times Blue + L)$ | [28] |
| Global vegetation index (GVI) | $-0.29 \times (G) - 0.56 \times (Red) + 0.6 (IR) + 0.49 (NIR)$ | [29] |
| Infrared percentage vegetation index (IPVI) | NIR/(NIR + Red) | [30] |
| Normalized difference vegetation index (NDVI) | (Red–NIR)/(Red + NIR) | [31] |
| Blue | Reflectance value of Landsat satellite band | Landsat satellite |
| Green | Reflectance value of Landsat satellite band | Landsat satellite |
| Red | Reflectance value of Landsat satellite band | Landsat satellite |
| Near-infrared (NIR) | Reflectance value of Landsat satellite band | Landsat satellite |
| Shortwave infrared (SWIR) | Reflectance value of Landsat satellite band | Landsat satellite |
| Principal components of Landsat bands | PC1, PC2, PC3, and PC4 | [32] |
| Normalized-NDVI | (NIR–(TM1 + Green))/(NIR + (TM1 + Green)) | [31] |
| Optimized soil-adjusted vegetation index (OSAVI) | (NIR–Red)/(NIR + Red + 0.16) | [33] |
| PD 311 | Red–TM1 | [34] |
| PD 312 | (Red–Blue)/(Red + TM1) | [34] |
| PD 321 | Red–Green | [34] |
| PD 322 | (Red–Green)/(Red + Green) | [34] |
| Ratio-Based | NIR/(Blue + Green) | [31] |
| Ratio vegetation index (RVI) | (NIR/Red) | [35] |
| Soil-adjusted vegetation index (SAVI) | $[NIR–Red)/(NIR + Red + L)] \times (1 + L)$ | [28] |
| Stress-related | $(TM1 \times Green)/Red$ | [31] |
| Transformed vegetation index (TVI) | (SWIR–Red)/(SWIR + Red) | [36] |
| VIT01 | Red/Thermal | [37] |
| VTI02 | Thermal/(Red + SWIR) | [37] |
| VIT03 | Thermal/Red | [37] |
| VIT04 | Thermal/(SWIR + Green) | [37] |
| Brightness index | $BI = ((Red \times Red) + (NIR \times NIR))^{0.5}$ | [38] |
| Normalized difference moisture index (NDMI) | $NDMI = (NIR - SWIR)/(NIR + SWIR)$ | [39] |
| Normalized difference snow index (NDSI) | $NDSI = (Red - NIR)/(Red + NIR)$ | [40] |
| Salinity index$_1$ (S$_1$) | S1 = Blue/Red | [41] |
| Salinity index$_2$ (S$_2$) | S2 = (Blue − Red)/(Blue + Red) | [42] |
| Salinity index$_3$ (S$_3$) | $S3 = (Green \times Red)/Blue$ | [42] |
| Salinity index$_4$ (S$_4$) | $S4 = (Blue \times Red)/Green$ | [42] |
| Salinity index$_5$ (S$_5$) | $S5 = (Red \times NIR)/Green$ | [42] |
| Salinity index$_6$ (S$_6$) | $S6 = (Blue \times Red)^{0.5}$ | [38] |
| Salinity index$_7$ (S$_7$) | $S7 = (Green \times Red)^{0.5}$ | [43] |
| Salinity index$_8$ (S$_8$) | $S8 = (Blue^2 \times Green^2 \times Red^2)^{0.5}$ | [43] |

In addition, Figure 3 shows examples of the spatial distribution of selected vegetation indices that were used to predict SOC for the study area. These indicies were calculated using the ArcGIS 10, Idrisi Selva, and ENVI 4.8 software. To evaluate the relationships

between remote sensing data and SOC, a Pearson correlation analysis was performed between SOC and each predictor.

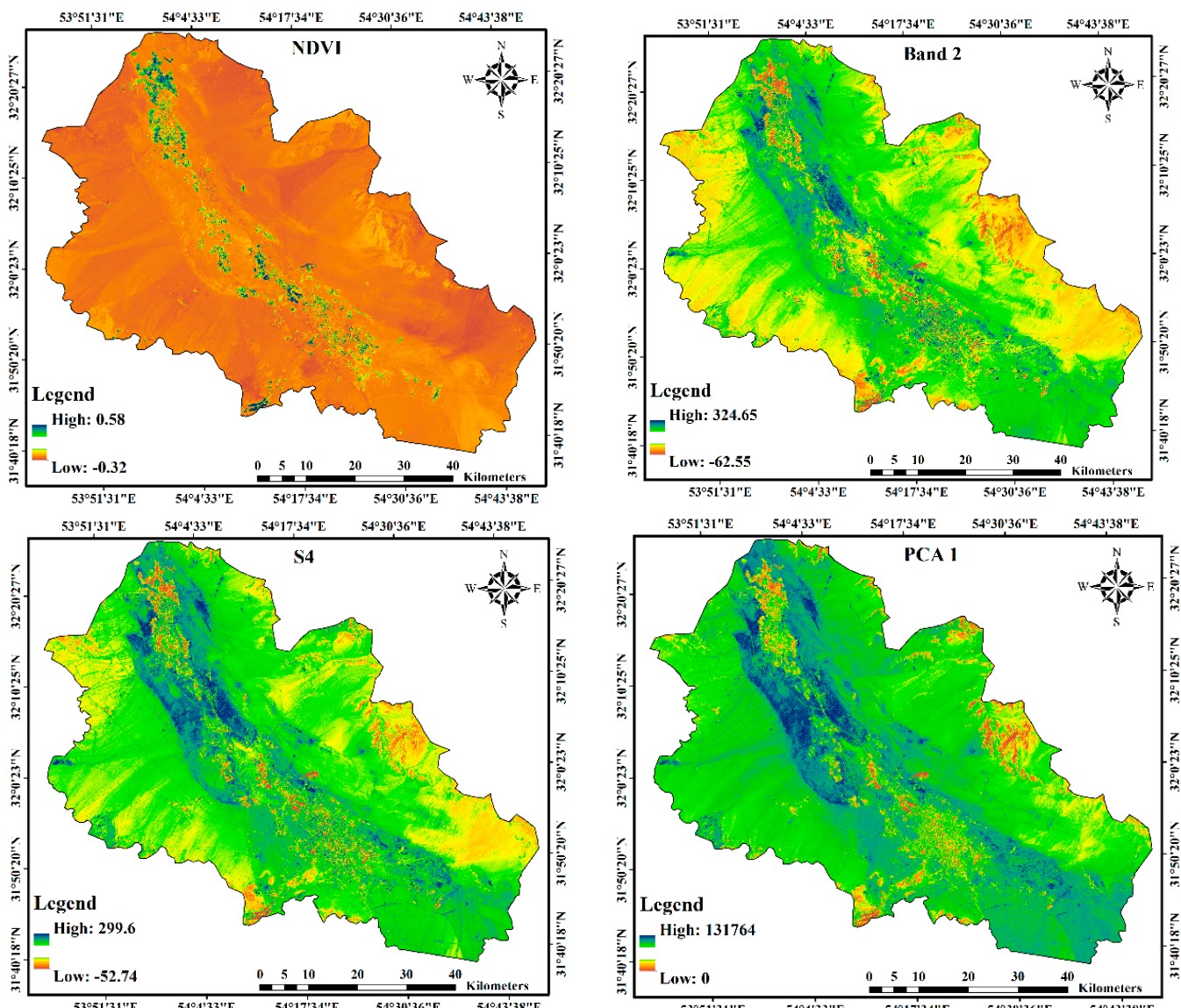

**Figure 3.** Map of environmental predictors used for predicting SOC in the Yazd-Ardakan Plain (see Table 1 for codes).

## 2.3. Machine Learning

In this study, the three ML models (RF, SVR, and ANN) were fitted and validated using the 2016 data. To evaluate the historical changes in the SOC, the best fitting ML model was applied to the remote sensing data collected for the periods 1986, 1999, and 2010.

### 2.3.1. Random Forest (RF)

Although a large variety of ML techniques have been tested in DSM [20], the RF learner [44] has recently seen great utility in Iran for predicting the spatial patterns of soils [20,24,45,46]. The RF learner is an example of an ensemble model, which is based on a unique set of CART-like decision tree models developed from a random bootstrap sample of the training dataset. The use of an ensemble model aims to minimize the effect of overfitting the model, which is a common problem with hierarchical, nonlinear models. Furthermore, additional randomness is introduced into the model whereby a random subset of the predictors is tested when generating the node-splitting rules at each node.

Here, the goal of the node-splitting rules is to maximize the uniformity within the nodes and the heterogeneity between the nodes with respect to the training data. When creating a set of individual decision trees, the trees are aggregated into a single predictive model. The main hyperparameters in the RF model include $n_{tree}$, which specifies the number of decision trees within the ensemble; and $m_{try}$, which specifies the number of predictors that are tested when generating each node-splitting rule. The default settings for these hyperparameters are $n_{tree} = 1000$ trees and $m_{try} = p^{0.5}$, where $p$ is the number of predictors. Based on these two parameters, decision trees are grown as large as possible and without pruning [20,25]. Since not all training observations are used to generate each decision tree, the out-of-bag samples can be used to perform a permutation-based, variable importance analysis and calculate the percent increase in mean square error (%IncMSE). A higher %IncMSE represents greater variable importance.

2.3.2. Support Vector Regression (SVR)

Support vector machines were proposed in the late 1960s by Vapnik and Lerner [47]. This supervised learning approach may be used for classification and regression purposes to perform dichotomy classification of multidimensional feature-vectors [47,48]. Support vector machines were originally created for classification purposes, where it seeks an optimal hyperplane to ensure the largest margin between classes, resulting in a higher likelihood of generalization. In regression (i.e., SVR), the model looks for a function that meets the error criteria, ignoring points near and distant from the decision boundary. These are the "low error" sites, with minimal residuals. Points outside the margin are permitted with a penalty weight. The penalty balances the effect of outliers by allowing points outside the regression function [48]. The ability of SVR to generalize is dependent on the tuning of hyperparameters.

2.3.3. Artificial Neural Networks (ANN)

Artificial neural networks (ANN) are highly adaptable computer networks that may be used to model complicated nonlinear interactions between variables [49]. The algorithm is based on a set of algorithm fitting functions that make no assumptions regarding error distribution [49,50]. When applied to large regions, it may provide the benefit of abstraction. An ANN model is developed in three stages: data production, optimum configuration selection, and validation on an independent data set.

*2.4. Accuracy and Uncertainty Assessment*

To evaluate the performance of the RF, SVR, and ANN models for SOC prediction, a 10-fold cross-validation procedure was used. The measures of accuracy used in this study included mean absolute error (*MAE*), coefficient of determination ($R^2$), and root mean square error (*RMSE*). These indicators are formulated as follows:

$$MAE = \frac{\sum_{i=1}^{n}|X_i - Y_i|^2}{n} \tag{1}$$

$$RMSE = \sqrt{\frac{1}{n}\sum_{i=1}^{n}(X_i - Y_i)^2} \tag{2}$$

$$R^2 = \frac{\sum_{i=1}^{n}\left(X_i^* - Y_i^*\right)^2}{\sum_{i=1}^{n}(X_i - Y_i)^2} \tag{3}$$

where $X_i$ and $Y_i$ correspond to the measured and predicted values, respectively; and $X_i^*$ and $Y_i^*$ correspond to the common of the measured and predicted values, respectively. The modeling and validation were carried out by using the *randomForest* and *caret* packages of the R 3.5.1 statistical software.

To assess the uncertainty of the three models, a leave-one-out cross-validation method was used. This method resulted in 201 predicted SOC maps. Based on the predicted maps,

the mean and standard deviation (SD) of SOC for each pixel were calculated. Then, the proportion of measured SOC that fell within the 90% prediction interval (i.e., prediction interval coverage probability; PICP) and mean prediction interval (MPI: upper prediction limit minus the lower prediction limit) were calculated to measure the quality of the uncertainty estimates.

## 3. Results and Discussion

### 3.1. Summary Statistics

A summary of the SOC data is shown in Figure 4. The minimum and maximum amount of SOC in the study area were 0.02% and 1.01%, respectively. Overall, the SOC contents had a mean value of 0.32% with a median of 0.28%, indicating that the study area had a low SOC. Furthermore, the low coefficient of variation of 1.24, indicates low spatial variability in SOC, with most of the study area having SOC values > 0.5%. These results were consistent with another study that found similarly low SOC in the Herat Plain in the Yazd Province of Iran [50].

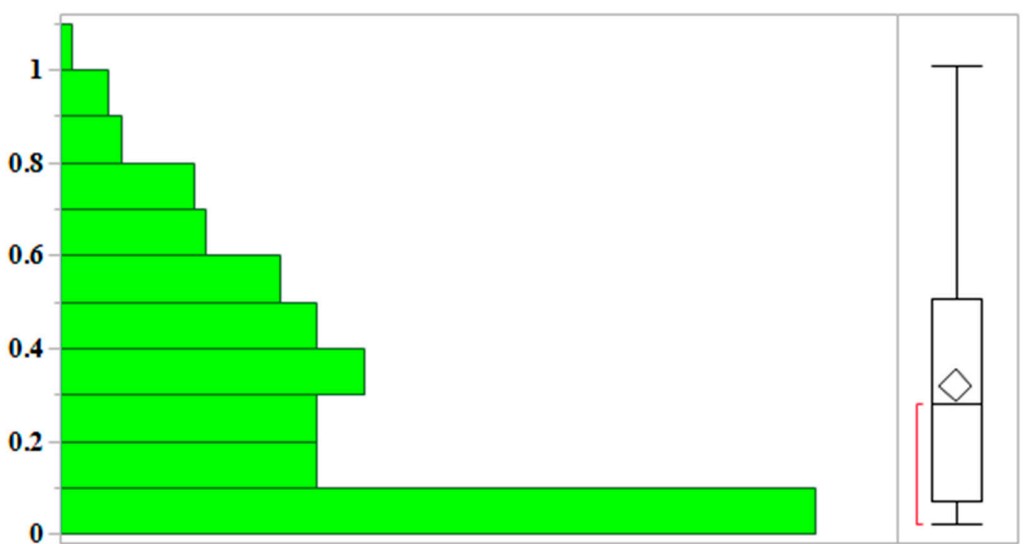

**Figure 4.** Distribution of SOC contents of the Yazd-Ardakan Plain.

The relationships between the remote sensing and SOC data were evaluated using Pearson's correlation coefficient analysis (Table 2). Here, 16 of the 37 environmental predictors were identified to be significantly correlated with SOC contents at *p* value < 0.01, while the other four predictors were significantly correlated at *p* value < 0.05. Several studies around the world have also reported a high correlation between SOC content and remote sensing predictors [13,50,51].

### 3.2. Accuracy and Uncertainty Assessments

The SVR, RF, and ANN models were tested using a 10-fold cross-validation process to model SOC. The evaluation results of the models are presented in Table 3. The results showed higher accuracy of the RF model ($R^2$ = 0.54; *RMSE* = 0.08%; *MAE* = 0.06) compared to SVR and ANN. Furthermore, we assessed the uncertainty of the models using PICP and MPI (Table 3). Theoretically, 90% of the observations should fall within the defined prediction interval with a confidence level of 90% while the MPI should be as narrow as possible. The results indicated that RF achieved the highest PICP (81%) and the lowest MPI (0.14) in predicting SOC, compared to the SVR and ANN models. Similarly, Pahlavan-Rad et al. [19] applied the RF model for SOC predictions for the low relief landscapes of eastern Iran and reported an *RMSE* = 0.16% and *MAE* = 0.21. Other studies have also described the effectiveness of RF learners as being dependable in predicting SOC and other soil properties [20,52]. Therefore, the RF model was selected for predicting the spatiotemporal patterns of SOC.

**Table 2.** Pearson's Correlation Coefficient (PCC) between environmental predictors and soil organic carbon content (codes refer to Table 1).

| Index | PCC | Index | PCC |
|---|---|---|---|
| CTVI | 0.37 ** | S6 | −0.09 |
| DVI | 0.38 ** | S7 | −0.07 |
| EVI | −0.37 ** | S8 | 0.27 ** |
| GVI | 0.33 ** | SAVI | 0.37 ** |
| IPVI | −0.11 | Stress-related | −0.04 |
| NDSI | −0.37 ** | TTVI | 0.37 ** |
| NDVI | 0.37 ** | TVI | 0.37 ** |
| NIR | 0.36 ** | VIT01 | −0.06 |
| OSAVI | 0.37 ** | VIT02 | −0.14 * |
| PD311 | −0.09 | VIT03 | 0.12 |
| PD312 | −0.12 | VIT04 | −0.08 |
| PD321 | −0.15 * | PCA1 | −0.16 * |
| PD322 | −0.20 ** | PCA2 | 0.03 |
| Ratio Based | −0.08 | PCA3 | −0.31 ** |
| RVI | −0.37 ** | PCA4 | 0.26 ** |
| S1 | 0.13 | Band1 | −0.11 |
| S2 | 0.12 | Band2 | −0.10 |
| S3 | −0.08 | Band3 | −0.08 |
| S4 | −0.14 * | Band4 | −0.10 |
| S5 | 0.07 | | |

** Correlation is significant at $p$ value < 0.01 (2-tailed). * Correlation is significant at $p$ value < 0.05 (2-tailed).

**Table 3.** Accuracy and uncertainty assessments of models to predict SOC.

| Model | $R^2$ | *RMSE* | *MAE* | MPI | PICP |
|---|---|---|---|---|---|
| RF | 0.536 | 0.082 | 0.058 | 0.14 | 81 |
| MLP | 0.501 | 0.142 | 0.110 | 0.19 | 75 |
| SVR | 0.407 | 0.181 | 0.134 | 0.21 | 67 |

*3.3. Variable Importance Analysis*

To further understand the soil-environmental relationships, variable importance analysis was carried out on the RF model (Figure 5). The analysis showed that NDVI was the most effective predictor of SOC with %IncMSE = 11.11%. In addition, CTVI, SAVI, and S3 were also highly ranked variables with a %IncMSE of 10.55%, 8.54%, and 6.77%, respectively. Similar studies showed that vegetation indices could increase the accuracy of modeling when predicting SOC in northern Iran [50]. In addition, the identification of NDVI as the most important predictor was also consistent with several other studies [46,50,53]. Similarly, Falahatkar et al. (2016) [54] also found that remote sensing indices are generally good predictors of SOC content, while Hong et al. (2002) [55] also showed that PCA2 and PCA4 were strongly correlated with soil chemical properties, such as soil organic matter. In contrast, Gomez et al. (2008) show that the SOC content was not related to the NDVI [56].

Because SOC directly influences soil color and its reflectance, remote sensing data, such as NDVI, CTVI and SAVI can characterize SOC variability, especially in undisturbed ecosystems [51]. Furthermore, soil salinity, which is influenced by agricultural expansion, land use, and the land cover type, can directly impact SOC input and turnover and may be characterized using the salinity indices used in this study. Thus, among the environmental covariates, three soil salinity indices (S3, S4, S5) were effective in explaining the variability of SOC.

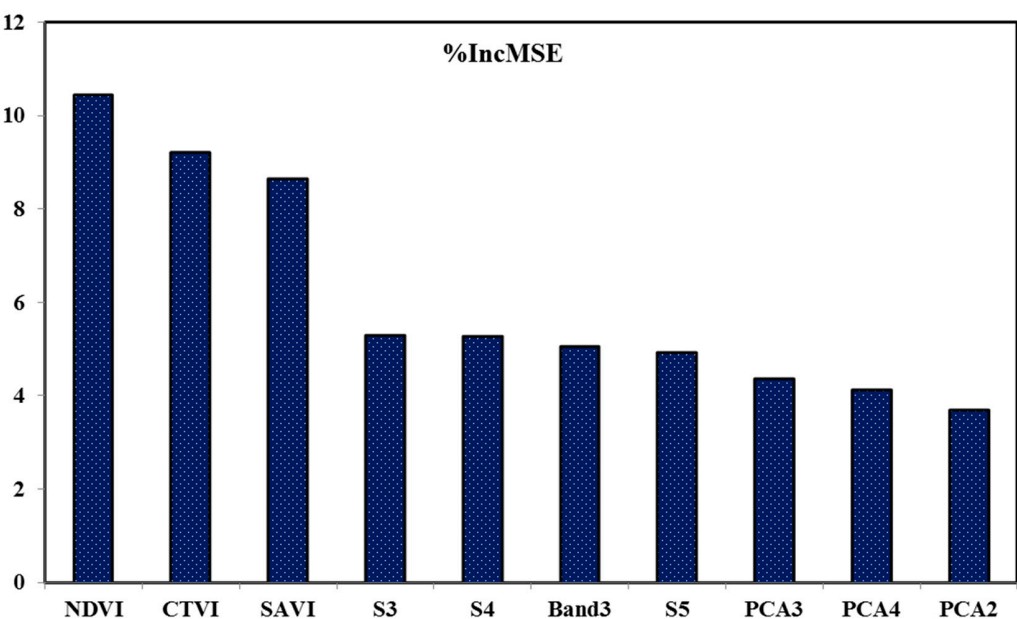

**Figure 5.** Variable importance analysis using the percent increase in mean square error (%IncMSE) for predicting soil organic carbon content using the Random Forest model.

### 3.4. Soil Organic Carbon Trends for 1986, 1999, 2010, and 2016

After selecting the fitted RF model as the best model, a SOC map was generated for 2016 at a 30 m spatial resolution for the Yazd-Ardakan Plain in Iran. Using historical remote sensing data, the RF model generated from the 2016 data was then applied to the 1986, 1999, and 2010 datasets (Figure 6). To evaluate the spatial-temporal changes in SOC and aid in interpreting the results, the SOC maps were reclassified into three classes in Figure 6, where SOC < 0.3%, SOC is between 0.3–0.6%, and SOC < 0.6%. When comparing the NDVI (Figure 3) and SOC map for 2016 (Figure 6), higher SOC corresponded with higher NDVI values. For example, NDVI values were as high as 0.58 in the central part of the study area, where there was abundant vegetation due to the presence of agricultural lands and orchards, and therefore, the highest SOC values were predicted in the same region. In comparison, the surrounding regions had substantially lower NDVI, with values reaching as low as −0.32; hence, the SOC values were low as well. Similar to Zhao et al. (2015), this study attributed the spatial patterns of SOC to the presence of croplands and agriculture, suggesting that land use is a key control of SOC [57].

To further examine the SOC changes across these four periods, the area of the SOC classes is presented in Figure 7. The results show that from 1986 to 2016, the classes with SOC content > 0.6% and 0.3–0.6% decreased by 25,888 ha (5.26%) and 138,272 ha (28.63%), respectively. In comparison, the area of the SOC < 0.3% class increased by 164,160 ha (33.99%) and thus indicating a general decrease in SOC from 1986 to 2016 (Figure 7). These changes could be linked to climate change, decreased groundwater quality, and decreased rainfall from 1986 to 2016. In addition, because of the increased soil salinity in the region, the cultivated lands also drastically decreased from 1986 to 2016. For example, Fathizad et al. (2020) reported that an increase in soil salinity was attributed to the expansion of agricultural lands, increased number of wells, and the overexploitation of groundwater resources [23]. Thus, agricultural activities in arid and semi-arid regions of Iran are the most important controls of SOC.

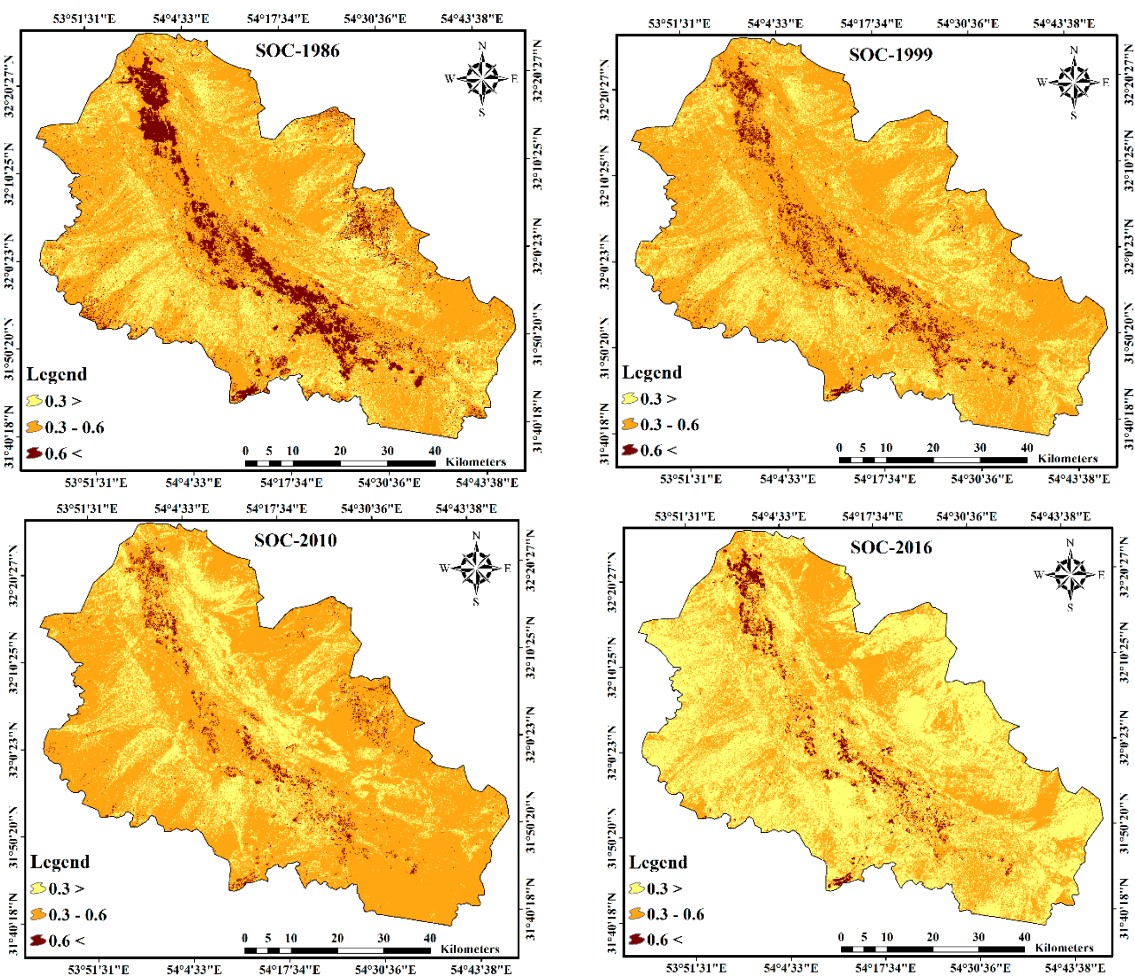

**Figure 6.** Predicted soil organic carbon (SOC) maps of the Yazd-Ardakan Plain for 1986, 1999, 2010, and 2016.

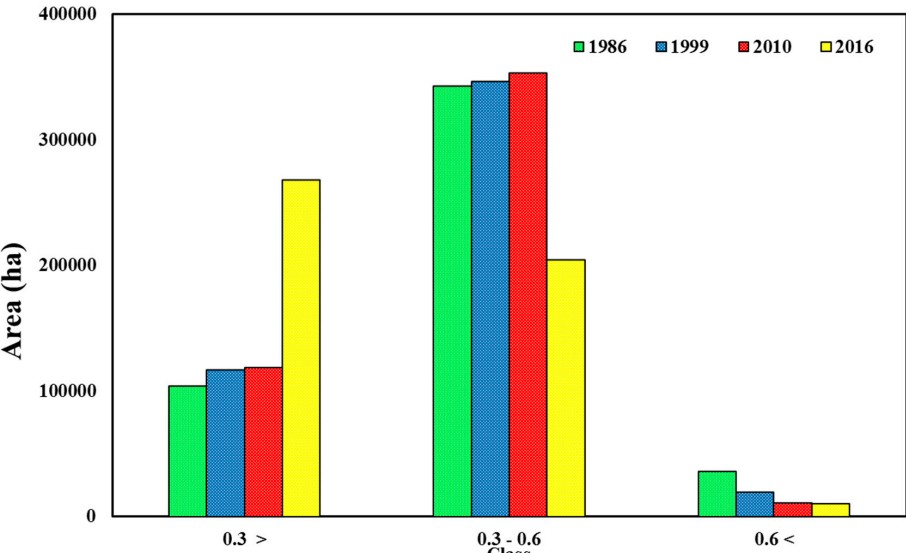

**Figure 7.** Areal extent of each soil organic carbon (SOC) class for the Yazd-Ardakan Plain for 1986, 1999, 2010, and 2016.

## 4. Conclusions

This study demonstrated the effectiveness of the RF model for predicting the spatiotemporal patterns of SOC content of the oasis and arid-agroecosystem area which the approach may be utilized in other similarly arid conditions. In general, the results showed that the RF model could be used for mapping the spatiotemporal dynamics of SOC content. The results revealed alarming changes in SOC content where areas with SOC = 0.3–0.6% decreased by 25,888 ha, and areas with SOC < 0.3% increased by 164,160 ha. These drastic changes resulted from reduction in agricultural activity and cultivatable areas from 1986 to 2016. During this period, the area of agriculture lands decreased by ~1.18%, while the areas of barren lands and sandy hills increased 5.16% and 0.09%, respectively. It can be concluded that the mismanagement of the lands, not only by the replacement of agricultural lands with residential areas, but also by declining water quality, as reported by Fathizad et al. [23], reduced agricultural activity and SOC. Soil dynamics, in addition to soil formation and evolution, are strongly influenced by soil management. Therefore, future research should be conducted with the focus of obtaining other environmental predictors to further investigate changes in other soil properties based on the use of DSM and machine-learning techniques. It is also suggested that RF models and other environmental predictors, such as land class and land cover data and satellite images with higher resolution be used in future studies. This study provides a method for understanding the spatiotemporal dynamics of SOC and the methods may be adapted to the monitoring of other soil properties.

**Author Contributions:** Conceptualization, H.F. and R.T.-M.; methodology, H.F. and R.T.-M.; software, H.F. and R.T.-M.; validation, H.F., R.T.-M., M.A.H.A., M.Z., B.H. and T.S.; formal analysis, H.F., R.T.-M. and M.A.H.A.; investigation, H.F., M.A.H.A., M.Z. and R.T.-M.; resources, H.F.; data curation, H.F. and M.A.H.A.; writing—original draft preparation, H.F., M.A.H.A., R.T.-M., M.Z., B.H. and T.S.; writing—review and editing, H.F., M.A.H.A., R.T.-M., M.Z., B.H. and T.S.; visualization, H.F. and R.T.-M.; supervision, M.A.H.A. and R.T.-M. All authors have read and agreed to the published version of the manuscript.

**Funding:** This research received no external funding.

**Institutional Review Board Statement:** Not applicable.

**Informed Consent Statement:** Not applicable.

**Acknowledgments:** Ruhollah Taghizadeh-Mehrjardi and Thomas Scholten have been supported by the Deutsche Forschungsgemeinschaft (DFG, German Research Foundation) under Germany's Excellence Strategy—EXC number 2064/1—Project number 390727645, and collaborative research center SFB 1070 'ResourceCultures'—Project number 215859406. Mojtaba Zeraatpisheh's postdoctoral program at Henan University, China, has been supported by the National Key Research and Development Program of China, grant numbers 2017YFA0604302 and 2018YFA0606500.

**Conflicts of Interest:** The authors declare no conflict of interest.

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
