# Peer review of "Spatiotemporal Assessment of Soil Organic Carbon Change Using Machine-Learning in Arid Regions"

_agronomy, doi:10.3390/agronomy12030628_

Round 1

Reviewer 1 Report

I have added some comments regarding wording and general merit of the proposed paper.

Author Response

Reviewer 1

Comments in the pdf:

Thanks for your comments. We modified all sections.

By sampling to 20 cm depth, you have effectively diluted the amount of variability in SOC that may have been observed in an area devoid of Signiant annual rainfall. Such areas are unlikely to have Signiant SOC below 10 cm depth, with very little between 10 and 20 cm. Budgets are always a consideration, but perhaps differentiating between these depths would be very useful, and helpful for correlating between the other parameters and SOC, i.e. it may be more sensitive. Additionally, the choice of a wet combustion method was appropriate given the likely lime contents of soils. The Walkley-Black method has several inconsistencies that may add to uncertainties in your results. In future, a Heanes type modification to the W-B method may give better (more consistent) results.

Many thanks for your comments. As you already mentioned, the budgets are always a consideration, However, for future works, we indeed take to account. Importantly, the method of analysis “Walkley-Black method” is a standard method to measure SOC in our country. But, this is a very important point you highlighted. It is worth mentioning that that might be one potential research: which method is better to quantify SOC in arid soils with calcareous parent materials. Many thanks. 

Reviewer 2 Report

In this study, the authors used remote sensing data to study the spatial and temporal change in soil organic carbon of a case study site. To my understanding, using algorithm to classify images is already well established, therefore, from the viewpoint of methodological improvement, the manuscript failed to provide any new insight. 

Author Response

Reviewer 2=

In this study, the authors used remote sensing data to study the spatial and temporal change in soil organic carbon of a case study site. To my understanding, using algorithm to classify images is already well established, therefore, from the viewpoint of methodological improvement, the manuscript failed to provide any new insight.

Many thanks for your comments. We should note that our work does not just image classification. We did digital soil mapping for the quantification of SOC in the study area. Added to this, we tried to model SOC in different time intervals. This information gives us the background about soil quality degradation in the time intervals and decreasing grassland areas in the study area.   

Reviewer 3 Report

The paper is well documented and the study is properly presented. The research area and methodology used is well presented. In my opinion, the presentation of the results can be improved. The quality of the graphics is inadequate, and the tables must contain an explanation of the abbreviations used in the footnote. The conclusions for future research in this field can also be improved.

Author Response

The paper is well documented and the study is properly presented. The research area and methodology used is well presented. In my opinion, the presentation of the results can be improved. The quality of the graphics is inadequate, and the tables must contain an explanation of the abbreviations used in the footnote. The conclusions for future research in this field can also be improved.

Many thanks for your comments. We improve the conclusion section.   

Reviewer 4 Report

The major comment regarding the result of random forest RF to retrieve SOC

The result was unconvincing to use remote sensing data  to retrieve soil organic carbon values, with a coefficient of determination (R2) of 0.53),

Therefore, the authors have to re-model the organic carbon using other methods such as support vector machine (SVM) or neural network (NN) 

The minor comments:

Language revision is required for all manuscripts where a few grammatical mistakes were noted

The methodology

More information should be given about random forest algorithms (RF) including the equations  

Flowchart of the work

Author Response

Reviewer 4

The major comment regarding the result of random forest RF to retrieve SOC. The result was unconvincing to use remote sensing data  to retrieve soil organic carbon values, with a coefficient of determination (R2) of 0.53), Therefore, the authors have to re-model the organic carbon using other methods such as support vector machine (SVM) or neural network (NN)

We appreciated the reviewer's concern about the accuracy. Random forest is the most common method for digital soil mapping. But we would like to note that in digital soil mapping these accuracies are common. Because soil variations especially SOC changes in the region depend on many factors. These changes can be captured by remote sensing data and terrain attributes perfectly. Although we used the other geospatial information, they also could not help to increase the accuracies significantly. However, we tested the other machine learning models, and the results are presented as an appendix.    

Language revision is required for all manuscripts where a few grammatical mistakes were noted

We improved the language.

More information should be given about random forest algorithms (RF) including the equations 

The equations are added.

Flowchart of the work

The flowchart is added.

Round 2

Reviewer 2 Report

As I stated in my previous version of review, the manuscript does not convey any significantly new findings. Rather, it is just an application of a well-established method to a new case study site. 

Author Response

As we stated in our previous version of the review, the paper contains the new findings. The approach applied in this study to reconstruct the past status of SOC is very interesting.

Reviewer 4 Report

the author/s did a great job, however, I have recommended the author/s to apply other models to predict the SOC and discuss in the result section, abstract and  methodology 
the comparison of the models is very important to be attractive in my point of view, otherwise, the manuscript still has weak

Author Response

Thanks for the comments. We added the results and methods of the other two ML models to the body of the manuscript.